



# The FluxEngine air-sea gas flux toolbox: simplified interface and extensions for *in situ* analyses and multiple sparingly soluble gases

Thomas Holding[1], Ian G. Ashton[1], Jamie D. Shutler[1], Peter E. Land[2], Philip D. Nightingale[2], Andrew P.
Rees[2], Ian Brown[2], Jean-Francois Piolle[3], Annette Kock[4], Hermann W Bange[4], David K. Woolf[5],
Lonneke Goddijn-Murphy[6], Ryan Pereira[7], Frederic Paul[3], Fanny Girard-Ardhuin[3], Bertrand Chapron[3],
Gregor Rehder[8], Fabrice Ardhuin[3], Craig J. Donlon[9]

[1]University of Exeter, Penryn Campus, Cornwall, TR10 9EZ, UK.
[2]Plymouth Marine Laboratory, Prospect Place, Plymouth, PL1 3DH, UK.
[3]Ifremer, Univ. Brest, CNRS, IRD, Laboratoire d'Oceanographie Physique et Spatiale (LOPS), IUEM,
Brest, France.
[4]GEOMAR Helmholtz Centre for Ocean Research Kiel, Marine Biogeochemistry Research Division,
24105 Kiel, Germany.
[5]International Centre for Island Technology, Heriot-Watt University, Stromness, Orkney, KW16 3AW,
UK.
[6]Environmental Research Institute, University of the Highlands and Islands, Thurso, KW14 7EE, UK
[7]The Lyell Centre, Heriot-Watt University, Research Avenue South, Edinburgh, EH14 4AS, UK.
[8]Leibniz-Institute for Baltic Sea Research Warnemünde, 18119 Rostock, Germany.
[9]European Space Agency, Noordwijk, The Netherlands.

Correspondence to: Thomas Holding (t.m.holding@exeter.ac.uk)

**Abstract.** The flow (flux) of climate critical gases, such as carbon dioxide ($CO_2$), between the ocean
and the atmosphere is a fundamental component of our climate and the biogeochemical development of
the oceans. Therefore, the accurate calculation of these air-sea gas fluxes is critical if we are to monitor
the health of our oceans and changes to our climate. FluxEngine is an open source software toolbox
that allows users to easily perform calculations of air-sea gas fluxes from model, *in-situ* and Earth
observation data. The original development and verification of the toolbox was described in a previous
publication and the toolbox is already being used by scientists across multiple disciplines. The toolbox
has now been considerably updated to allow its use as a Python library, to enable simplified
installation, verification of its installation, to enable the handling of multiple sparingly soluble gases
and greatly expanded functionality for supporting *in situ* dataset analyses. This new functionality for
supporting *in situ* analyses includes user defined grids, time periods and projections, the ability to re-
analyse *in situ* $CO_2$ data to a common temperature dataset and the ability to easily calculate gas fluxes
using *in situ* data from drifting buoys, fixed moorings and research cruises. Here we describe these new
capabilities and then demonstrate their application through illustrative case studies. The first case study
demonstrates the workflow for accurately calculating $CO_2$ fluxes using *in situ* data from four research
cruises from the Surface Ocean $CO_2$ Atlas (SOCAT) database. The second case study shows that
reanalysing an eight month time series of $pCO_2$ data collected from a fixed station in the Baltic Sea can
remove errors equal to 35% of the net air-sea gas flux. The third case study demonstrates that



biological surfactants could supress individual nitrous oxide sea-air gas fluxes by up to 13%. The final case study illustrates how a dissipation-based gas transfer parameterisation can be implemented and used. The updated version of the toolbox (version 3) and all documentation is now freely available.

## 1. Introduction

The exchange of climate relevant gases between the oceans and atmosphere including that of carbon dioxide ($CO_2$), nitrous oxide ($N_2O$) and methane ($CH_4$) is a major component of the climate system, and the ability of the oceans to absorb and desorb these gases varies both temporally and spatially. The need to monitor this exchange has been the driver for international data collation initiatives such as the Surface Ocean $CO_2$ ATlas (SOCAT, (Bakker *et al.*, 2016)) and the MarinE MethanE and NiTrous Oxide database (MEMENTO, Kock and Bange, 2015). These collaborative efforts are now routinely collecting, quality controlling and collating over a million new *in situ* data points each year. FluxEngine complements these initiatives by providing a standardised tool, which can robustly calculate air-sea gas fluxes from such *in situ* data, with the flexibility to incorporate new data sources and methodologies. The use of common tools and methods simplifies collaborations and accelerates advancements, both within and between scientific disciplines, through eliminating methodological or implementation-driven differences and the duplication of effort.

### 1.0 Overview of FluxEngine

FluxEngine is a flexible open source toolbox that allows users to easily exploit Earth observation and model data, in combination with *in situ* data, to calculate air-sea gas fluxes (Shutler *et al.*, 2016). The toolbox uses plain text-format configuration files allowing the user to configure the input data sources, the temporal period for the analysis, the structure of the air-sea gas flux calculation and user-defined gas transfer velocity parameterisations. Further optional features include the addition of random noise or bias to the input data. The calculation itself can be performed using fugacity, partial pressure or concentration data, using a bulk formulation or more accurate formulations that take into account vertical temperature gradients across the mass boundary layer, the very small layer at the surface over which gas exchange occurs. The latter approach allows a more accurate gas flux calculation and is described in detail by (Woolf *et al.*, 2016) and takes the generalised form of

$$F = k(\alpha_W \, fG_W - \alpha_S \, fG_A) \qquad (1)$$

where F is the sea-to-air flux of a sparingly soluble gas G, *k* is the gas transfer velocity (cm h$^{-1}$), $\alpha_S$ and $\alpha_W$ are the solubilities of the gas above and below the surface water interface and $fG_A$ and $fG_W$ are the respective fugacities. Here we use 'p' and 'f' prefixes to refer to partial pressure and fugacity of a gas, respectively. Gas transfer velocity is driven by turbulence at ocean surface, caused by wind stress and wave breaking, amongst other processes. Because of the wide availability of high quality wind data products and the relative difficulty of directly measuring turbulence, it is commonplace to estimate k using a statistical relationship with wind speed, e.g. (Ho *et al.*, 2006; Nightingale *et al.*, 2000; Wanninkhof, 2014).


Concentration of the gas is determined by its solubility and fugacity (or partial pressure). Equation (1) can therefore be rewritten as a product of the gas transfer velocity and the difference in gas concentrations,

$$F = k(G_W - G_S) \qquad (2)$$


where $G_S$ and $G_W$ are the concentration of the gas at and below the interface. The FluxEngine configuration file allows users to choose the structure of the gas flux calculation (i.e. a bulk calculation, or equation 1 or 2), the inputs and the gas transfer velocity (either by choosing an already implemented published algorithm or through parameterising their own). The user can then run all calculations across

their chosen input data and the outputs are Climate Forecast (CF) standard netCDF 4.0 files that contain data layers for each of the stages of the calculation, along with process indicator layers to aid the understanding of the calculated gas fluxes (such as surface chlorophyll-a concentrations, the climatological position of temperature fronts and error indicator layers).

Version 1.0 of FluxEngine was introduced and described by (Shutler, *et al.*, 2016), which included a full description of the calculations, the flexibility of the toolbox, and the extensive verification of the different calculations along with examples of its use. Since its original release the toolbox has continued to be developed and extended based on feedback from the user communities and the needs of specific scientific studies (e.g. Ashton *et al.*, 2016). These developments have considerably extended

the functionality of the toolbox and broadened the range of possible applications to which it can be applied. At the time of writing the toolbox and resulting data have been used to quantify regional method uncertainties (e.g. Wrobel and Piskozub, 2016; Wrobel, 2017), evaluate the impact of gas transfer processes on regional and global gas exchange (e.g. Ashton *et al.*, 2016; Pereira *et al.*, 2018), evaluate the European shelf sea $CO_2$ gas-fluxes and sink (Shutler *et al.*, 2016) and investigate

biological and physical controls of air-sea exchange (Henson *et al.*, 2018). FluxEngine has also been used to identify shortfalls of current modelling approaches through the inclusion of FluxEngine outputs within an international inter-comparison (Rödenbeck *et al.*, 2015) and is currently being used within two pan-European carbon monitoring research infrastructure projects (EU RINGO and EU BONUS INTEGRAL) which are part of the Integrated Carbon Observing System, ICOS. The toolbox has also

been incorporated within undergraduate and postgraduate teaching (e.g. at the University of Exeter within geography, environmental science and marine biology degrees, and at Utrecht University for computer science). Most recently the toolbox is being used by two European Space Agency (ESA) projects to support preliminary studies for a new satellite concept (the Sea Surface Kinematics Multiscale Monitoring, SKIM, satellite mission (Ardhuin *et al.*, 2018)) and to verify our understanding

of vertical temperature profiles and concentration gradients (as described by Woolf *et al.*, 2016) through the analysis of a novel fiducial reference dataset. The results from these studies will be reported elsewhere, but their needs have driven some of the advancements presented here.

This paper uses four case studies to illustrate key developments and extended capabilities now

contained within version 3.0 of the FluxEngine toolbox. Collectively the case studies illustrate user selectable grids, support for calculating sea-to-air gas fluxes from time series data collected by fixed monitoring stations and research cruises (and how to incorporate the flux outputs into the original dataset to create a coherent time series), the ability to calculate nitrous oxide ($N_2O$) and methane ($CH_4$)




sea-to-air gas fluxes, the addition of a new forcing variable (kinetic energy dissipation rate) and the
        ability to run ensembles of any of these calculations to characterise method uncertainties. The extensive
        support for *in situ* data contained within version 3 of FluxEngine means that it can now be fully
        exploited by three different scientific communities in isolation: *in situ*, model and Earth observation;
        whilst the original capability to enable gas fluxes to be calculated from combinations of *in situ*, model
        and Earth observation data is retained.


        Section 2 describes the structural extensions and changes, including the automatic software installers
        and verification tools (allowing users to verify the integrity of their installation). It explains how the
        toolbox can now be used as a command line tool or as a Python library. Section 3 then presents the case
        studies, while section 4 outlines the future direction and developments for the toolbox and section 5
gives conclusions. To aid the user the Appendices of this paper provide a list all of the toolbox utilities
        (Sect. 6) and details of all data sets used (Sect. 7).

        **2. New capabilities**

        The following sections describe the extensions to the FluxEngine toolbox that are now contained within
version 3.

        **2.1. Installation, verification and use**

        FluxEngine has now been optimised for use on a standalone desktop or laptop computer, removing the
        previous requirement for specialist computing facilities. Installation tools or instructions are now
provided for the following operating systems: Ubuntu/Debian based Linux
        (*install_dependendies_ubuntu.sh*), Apple Mac (*install_dependencies_macos.sh*) and Windows
        (instructions are within *FluxEngineV3_instructions.pdf*). Separate utilities (*verify_takahashi09.py* and
        *verify_socatv4_sst_salinity_gradients_N00.py*) can then be used to verify that FluxEngine has been
        successfully installed. These verification utilities run standard global sea-to-air $CO_2$ gas flux
calculations and net integrated fluxes using the (Takahashi *et al.*, 2009) sea-to-air $CO_2$ flux climatology
        (for year 2000) and the Woolf *et al.,* (in-review) Surface Ocean $CO_2$ Atlas (SOCAT, Bakker *et al.*,
        2016) derived sea-to-air $CO_2$ flux reference dataset (for year 2010). The results are then evaluated
        against the published reference data provided by Holding *et al.*, (2018) and the installation is deemed
        successful if all results are identical to the reference dataset within a precision of 5 decimal places. An
additional utility (*run_full_verification.py*) enables the user to perform a more detailed verification
        against both of these climatologies by executing a suite of 12 different configurations and scenarios, the
        justification for which are described within Woolf *et al.*, (in-review). Owing to the large volume of data
        required to execute and verify all of these scenarios, the verification data are not packaged with the
        standard FluxEngine download, but are all freely available and contained within Holding *et al.*, (2018).


        FluxEngine is now implemented as a Python module available on a creative commons license via
        http://github.com/oceanflux-ghg/FluxEngine. This means that FluxEngine and its accompanying
        utilities can be used as command line tools (stand-alone tools or called from another piece of software)
        or imported as a Python module and easily integrated with other software. This approach offers a larger
degree of flexibility than offered by version 1 of the toolbox and supports advanced exploitation. For
        example, a simple Python script can be written to run a sensitivity analysis where ensembles of flux
        calculations are required without any need to modify the underlying FluxEngine software.


To provide an indication of the execution time a benchmarking analysis was performed using an Intel Core i5 5.7 GHz Laptop processor with 8GB RAM running MacOS El Capitan. The automatic installation took ~3 minutes to complete and the basic verification script using the Woolf *et al*., (in-review) reference dataset (involving a global one year analysis of the gas fluxes for 2010, monthly temporal resolution and $1° \times 1°$ spatial resolution) took approximately 6 minutes to complete. As the flux calculation is sequential for each grid cell the execution time scales approximately linearly with number of grid points and number of time steps. Hence, doubling the temporal resolution will approximately double the execution time, whilst doubling the resolution of both spatial dimensions will lead to a factor of four increase in execution time.

### 2.2. Flexible input data specification

Previous versions of FluxEngine required the user to make changes to the underlying software in order to use new or differently formatted sources of input data. This required additional (and time consuming) testing and verification after modifications were made, making FluxEngine less accessible to those unfamiliar to Python programming. Two features have been added in version 3.0 to address this issue: i) file pattern matching (through standard Unix glob patterns and custom date/time tokens, described fully in *FluxEngineV3_instructions.pdf*) allows input file name format and directory structure to be customised using the plain text configuration file, ii) optional pre-processing functions can be used to manipulate input data after the data have been read into memory. These features can be specified for each input variable in the configuration file and FluxEngine contains a selection of common pre-processing functions, such as unit conversions or matrix transformation of the input data. Additional custom pre-processing functions can be added and tested easily by the user without the need to modify the core FluxEngine software, through copying and then completing the Python template function provided within the source code (*data_preprocessing.py*). Storing the completed function into the *data_preprocessing.py* file will then result in the custom pre-processing function being automatically available for use in any configuration files.

These features make it possible to use any observational netCDF dataset by specifying the file path and, if required, appropriate pre-processing functions. For example a custom pre-processing function could resample the input files, followed by a transformation to change the projection. This flexibility is conceptualised by the diagram in Fig. 1.

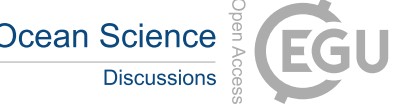

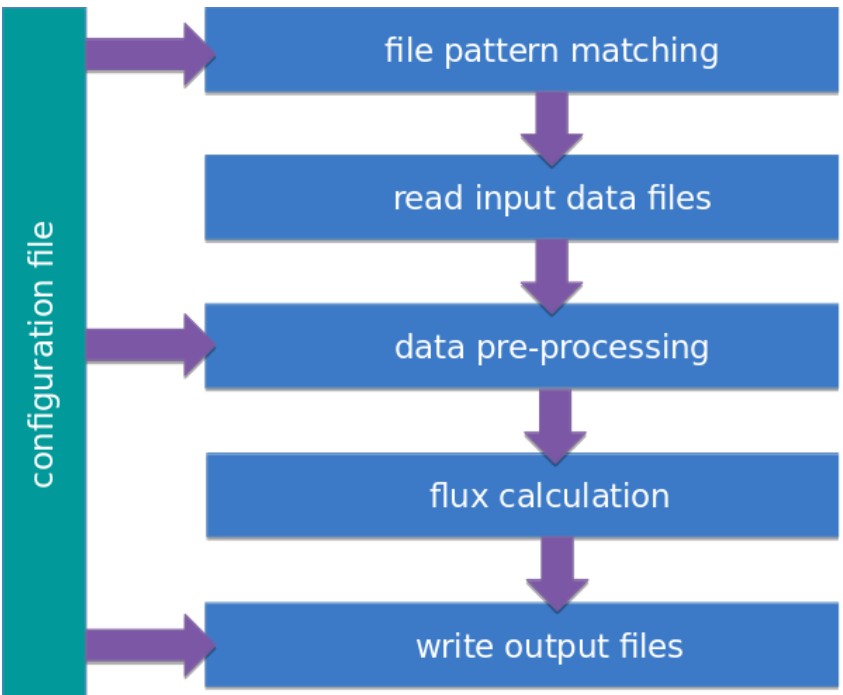

**Figure 1: Conceptual diagram showing the way that input data are imported and used by FluxEngine. Single or groups of files are specified using a plain text configuration file. File names are interpreted using a subset of regular expression matching syntax (Unix glob patterns) and additional tokens are used to substitute time and date. The data pre-processing steps occur after input files are read into memory. Pre-processing functions are specified in the configuration file. Finally, the netCDF output files follow a user-specified filename and directory structure (as specified in the configuration file).**

### 2.3. Extensive support for *in situ* data analyses

Version 1 of FluxEngine required that all input data be supplied as monthly 1° × 1° global grids. These

constraints precluded its application to regional analyses and *in situ* analyses, where sub-daily or sub-km resolutions are often more appropriate. The spatial resolution and extent can now be fully specified by the user and regional masks can be used in conjunction with the *ofluxghg_flux_budgets.py* tool to calculate regional net integrated fluxes. In addition, flexible start and stop times and user-specified temporal resolution allows gas fluxes to be calculated for specific time intervals, e.g. the calculation

can be configured to match the temporal resolution of the *in situ* data. Furthermore, a new configuration option allows output from multiple time points to be grouped into a single netCDF file (rather than multiple files, one for each time point). This feature is designed to enable the calculation of gas fluxes from fixed research stations and other scenarios in which it is more convenient to provide results as a single time-series.


Improvements have been made to the bundled file conversion utilities, which convert between plain text data formats and the netCDF format used by FluxEngine. By default, these tools use the SOCAT format (Bakker *et al.*, 2016) for convenience, but now offer a high degree of flexibility to reflect the variety of data formats and conventions used for storing *in situ* data. This means that the tools can be


used with virtually any text formatted *in situ* data files, avoiding the need for the user to convert their data to a fixed format with predefined column names.

The new utility, *append2insitu.py*, is designed specifically for use with *in situ* data and appends FluxEngine output as new columns within the original *in situ* data (achieved by matching spatial and temporal coordinates). For example, this means that users can use SOCAT (or custom) formatted *in situ* data as input to FluxEngine and then the results can be placed into a copy of the original input file, allowing the user to study the calculated fluxes, gas transfer rates, gas concentrations etc. alongside (and aligned with) their original *in situ* data. This functionality is demonstrated in case studies one and three within this paper.


*In situ* $fCO_2$ measurements are often made using water sampled from differing depths and/or a range of different instrument setups. A second new utility, *reanalyse_socat_driver.py* enables $fCO_2$ measurements to be re-analysed to a consistent temperature field at a consistent depth. This reanalysis tool is $CO_2$ specific and is required for an accurate gas flux calculation as it allows the *in situ* gas concentration to then be calculated at the bottom or top of the mass boundary layer, rather than assuming that the gas concentration at some depth is representative of that at the sea surface (Woolf *et al*., 2016). This reanalysis is especially important if the *in situ* data consist of a collated dataset originating from multiple instruments, sampling strategies or sources. In this situation the *in situ* measurements are more likely to be collected from a range of different depths. It is worth noting that ship draught, and thus underway measurement intake depth, can even vary on a single vessel due to changes in sea state, ballasting or cargo. A more detailed justification of the method and a full description of the approach are described in Goddijn-Murphy *et al.*, (2015). Whilst the reanalysis method and utility is $CO_2$ specific, its applicability to alternative gases (including unreactive $N_2O$ and $CH_4$) is discussed and shown in Table 1 of (Woolf *et al*., 2016). The impact of not performing this reanalysis on a relatively large time series of $CO_2$ measurements through the north and south Atlantic is demonstrated within case study one.

A typical workflow for calculating sea-to-air gas fluxes from *in situ* data using FluxEngine, and the tools used at each step, is illustrated in Fig. 2. All of the *in situ* analysis utilities, including the use of the *reanalyse_socat_driver.py* tool, are demonstrated in case studies one to three (Sect. 3).





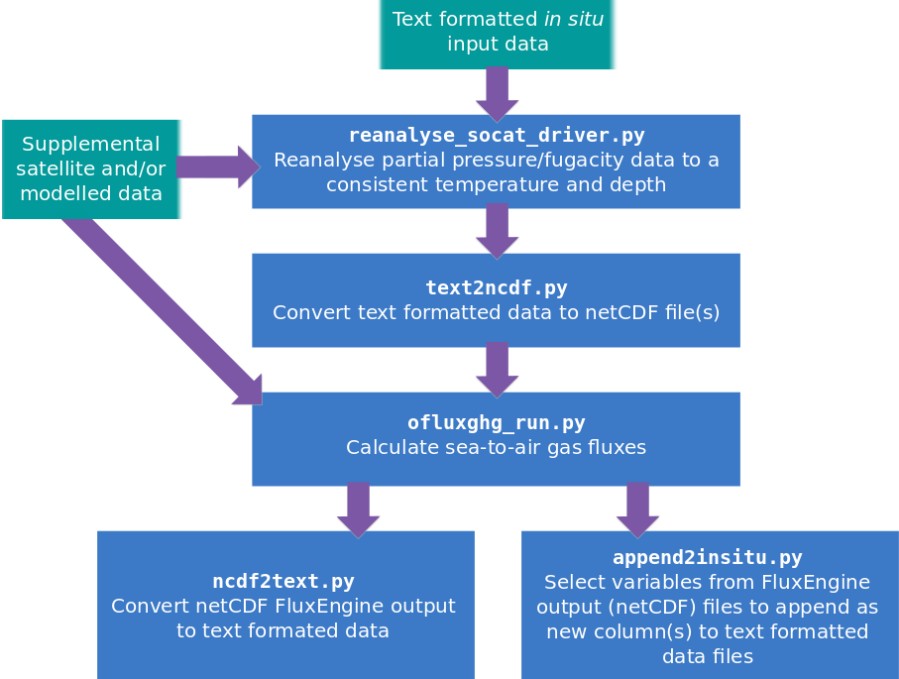

**Figure 2: A typical CO₂ workflow for using FluxEngine with *in situ* data, showing the different utilities (blue boxes) and input data (green boxes) used at each stage.**

### 2.4. Custom gas transfer velocity parameterisation

The processes that govern exchange, their relative importance and how gas exchange should be parameterised are all active areas of research. For example, a recent comparative study using FluxEngine highlighted a difference of up to 65% in global net $CO_2$ flux caused simply by using different wind-based gas transfer velocities (Wrobel and Piskozub, 2016).

FluxEngine has always allowed users to select or define different (mostly wind-based) transfer velocity parameterisations. However, version 3.0 adopts a modular approach to specifying the flux calculation, which makes it simpler for the user to extend the functionality and incorporate new gas transfer parameterisations. Custom parameterisations can be implemented as separate Python classes without modifying the core FluxEngine software. This is achieved by copying and modifying the template class provided in *rate_parameterisation.py*. Storing the new class within *rate_parameterisation.py* means that the new parameterisation will be automatically incorporated into the toolkit ready to be selected in the configuration file for all users of the particular FluxEngine installation. These custom parameterisations can define new input variables and therefore make use of additional input data files that can be included in the configuration file and will be automatically loaded into memory without requiring any additional setup. These custom parameterisations can also produce new data layers in the final netCDF output, such as the results from intermediate calculation steps, which may be useful for testing or subsequent analysis outside of FluxEngine. Examples of how to use this functionality are provided in the source code. The toolbox documentation describes the process of, and best practices for, extending FluxEngine in this way (see Sect. 9.1 and 9.2 within *FluxEngineV3_instructions.pdf*).



This increased flexibility means that users can define and use region-specific gas transfer parameterisations or incorporate new transfer processes into existing gas transfer parameterisations (such as the impact of biological surfactants as discussed by Pereira *et al.*, 2018). Case studies one (Sect. 3.1) and two (Sect. 3.2) demonstrate the use of different gas transfer parameterisations, while case study three (Sect. 3.3) demonstrates the use of a custom gas transfer velocity parameterisation,

which is used to assess the impact of biological surfactants on the $N_2O$ gas fluxes. Case study four (Sect 3.4) utilises a gas transfer velocity parameterisation that uses turbulent kinetic energy dissipation and provides an example of using additional input data.

### 2.5. Extensions for other sparingly soluble gases

The toolbox now supports the handling of two other sparingly soluble gases, ($CH_4$ and $N_2O$), and so gas specific data can be substituted into Eq. (1) or Eq. (2) (dependent upon the choice of setup). Gas specific parameterisations for Schmidt number (Sc) and solubility (α) are automatically chosen from those provided in Wanninkhof, 2014. The option to use the older Sc and solubility parameterisations from Wanninkhof, 1992 is also included for compatibility with previous versions and to aid

comparative analysis. It is worth noting that both sets of Sc parameterisations are only valid for salt water (35 PSU), and care should be taken when using them for analysis of freshwater data, or regions with lower salinity (e.g. the Baltic Sea, see case study two, Sect. 3.2). Support for additional and user-defined Schmidt number parameterisations are likely to be added in the future. FluxEngine can calculate dissolved gas concentration from the gas input data, which can be supplied as either partial

pressure or mean molar fraction of a gas in the dry atmosphere. Alternatively, dissolved gas concentrations can be provided directly as an input.

### 3. Case study examples of the new capabilities

The following sections describe the application and results from four case studies that illustrate the new

capabilities. Table 1 summarises the new features that are demonstrated in each case study. The respective configuration file for each case study can be accessed via the FluxEngine GitHub repository (http://github.com/oceanflux-ghg/FluxEngine).

| | New features utilised |
|---|---|
| **Case study 1:** Calculating sea to air $CO_2$ gas fluxes from research cruise data | Flexible input data specification to select *in situ* data files and unit conversion using pre-processing functions (Sect. 2.2). <br><br> Utilises new support for *in situ* data analysis, including the use of the *text2ncdf.py* and *append2insitu.py* tools, custom temporal resolution, reanalysis of fCO2 to a consistent temperature field. |
| **Case study 2:** calculating sea to air $CO_2$ gas fluxes from Östergarnsholm fixed station data. | Flexible input data specification to select *in situ* data files and unit conversion using pre-processing functions (Sect. 2.2). |



| | |
|---|---|
| | Utilises new support for *in situ* data analysis, including use of *text2ncdf.py*, daily temporal resolution, use of the reanalysis tool and output formatted as time-series (Sect. 2.3). |
| **Case study 3:** Surfactant suppression of sea to air $N_2O$ gas fluxes using the MEMENTO database. | Flexible input data specification and unit conversion using pre-processing functions (Sect 2.2). Utilises new support for *in situ* data analysis, including use of the *text2netcdf.py* and *append2insit.py* tools, custom temporal resolution and cruise-specific time interval (Sect. 2.3). Custom gas transfer parameterisation (Sect. 2.4). Calculation of $N_2O$ gas fluxes (Sect. 2.5). |
| **Case study 4:** Gas transfer velocity parameterisation using turbulent kinetic energy dissipation rate. | Unit conversion and use of custom pre-processing functions to calculate the dissipation rate from the input data. This uses the pre-processing functions to perform a non-trivial computation (Sect. 2.2). Use of a custom gas transfer parameterisation which includes the specification of an additional input data layer (Sect. 2.4). |

Table 1: Summary of the new functionality demonstrated in each research case study.


### 3.1 Case study 1: Calculating CO₂ fluxes from research cruise data

Each year over 1 million new *in situ* data points are included within the annual updates to the SOCAT dataset. Field scientists collecting these data often need to calculate the coincident sea-to-air gas fluxes, either using solely *in situ* measurements or through combining them with satellite Earth observation

and/or model data.

Here we illustrate the procedure for calculating sea-to-air gas fluxes from *in situ* data collected during four different sampling campaigns. These *in situ* data (Kitidis and Brown, 2017; Schuster, 2016; Steinhoff *et al.*, 2016; Wanninkhof *et al.*, 2016) were all collected in the north Atlantic during October

2013. For convenience these are referred to as cruises 1-4, respectively. The *in situ* data were first downloaded from PANGAEA (an open access data publishing and archiving repository) in tab-delimited format. The datasets follow the standard SOCAT structure and content (see Bakker *et al.*, 2016 table 9) and so they include sea surface temperature, salinity, surface air pressure, and fugacity of $CO_2$ in the seawater ($fCO_2$).


The majority of the measurements needed for the sea-to-air $CO_2$ gas flux calculation were measured *in situ* and exist within the downloaded datasets. However, wind speed (for the gas transfer parameterisation) was missing in all cases. Therefore to complement these *in situ* data, multi-sensor merged wind speed data at 10 m were downloaded (Cross-Calibrated Multi-Platform, CCMPv2, 6 hour

temporal resolution, $0.25^{\circ} \times 0.25^{\circ}$ spatial grid (Atlas, *et al.*, 2011)). These wind speed data were appended to the *in situ* data by matching each *in situ* measurement to the closest temporal and spatial grid point. This same process was used to add columns for the second and third moments of wind speed, which were estimated by taking the second and third power of wind speed, respectively.

Two datasets (Schuster, 2016; Steinhoff, *et al.*, 2016) were missing molar fraction of $CO_2$ in dry air ($xCO_2$) data, and so the same method of matching temporal and spatial grid points was used to fill in these fields using the GLOBALVIEW $CO_2$ dataset from the US National Oceanic and Atmospheric Administration (NOAA) Earth System Research Laboratory (ESRL) (GLOBALVIEW-CO2, 2013). For ease, these additional wind speed and $xCO_2$ data were downloaded, extracted and then inserted into

the tab delimited *in situ* file using some simple custom python scripts but the same process could be performed manually. These scripts are not part of FluxEngine but the functionality they provide will likely be available as part of the planned interactive Jupyter tutorials, see Sect. 4.

The *in situ* data were collected from different ships and underway systems, all sampling water at
different and unknown depths. These measurements are typically collected from a few metres below the water surface, whereas the $CO_2$ concentration (combination of $fCO_2$ and solubility) either side of the mass boundary layer is required for an accurate gas flux calculation. Before these data from multiple sources can be used for an accurate gas flux calculation, they need to be reanalysed to a common temperature dataset and depth (Goddijn-Murphy *et al.*, 2015; Woolf *et al.*, 2016). Therefore
the *reanalyse_socat_driver.py* tool was first used to reanalyse all $fCO_2$ data to a consistent temperature and depth.

In the absence of coincident *in situ* skin (or sub-skin) temperature data, the slow re-equilibrium time of $CO_2$ in seawater (i.e. on the order of months for $CO_2$ to equilibrate with the atmosphere) ensure that
monthly mean, or rolling monthly mean (centred on the day of interest) skin or sub-skin sea surface temperature (SST) values are suitable for re-analysing the *in situ* data. Arguably a robust daily skin or sub-skin SST value would be better, even if that is obtained by a seasonal curve fitted to the monthly values and interpolated to the day of interest. Here for simplicity monthly mean sea surface temperatures from the Reynolds Optimally Interpolated Sea Surface Temperature dataset (OISST,
Reynolds *et al.*, 2007) were used as the reference subskin temperature dataset, resulting in reanalysed $fCO_2$ that are valid for the bottom of the mass boundary layer (termed sub-skin within Woolf *et al.*, 2016).

The reanalysed $fCO_2$ were then inserted into the tab-delimited *in situ* dataset producing a single dataset.
The tab-delimited file was then converted into a netCDF format file using the *text2ncdf.py* tool. This tool groups all data according to a user-specified spatial sampling grid, calculating the mean value and standard deviation for each cell within the grid as well as the number of data that were used to calculate these statistics. Here, for simplicity, the spatial resolution was defined as $1^{\circ} \times 1^{\circ}$ grid. FluxEngine was





then configured to use each of the variables in the resulting netCDF file as input, with a pre-processing function applied to convert Reynolds OISST from Celsius to Kelvin (as all SST data within the main flux calculation use Kelvin). In order to produce a single netCDF output file for the entire 35 day period the temporal resolution for the flux calculation was set to 35 days. This allows the cruise tracks from all four cruises (1-4) to be easily visualised at the same time.

The sea-to-air $CO_2$ fluxes were then calculated using the rapid model (see Eq. (1) and Woolf *et al.*, 2016) and was run using a quadratic wind speed based gas transfer velocity parameterisation (Ho *et al.*, 2007). To identify the impact of the $fCO_2$ reanalysis stage, the sea-to-air $CO_2$ flux calculation was repeated using the original *in situ* $fCO_2$.

Figure 3a shows the resultant calculated $CO_2$ flux along each of the cruises (1-4). The southern sub-tropical part of the cruise track 1 represents an area of the ocean that is a sink of $CO_2$ (negative sea-to-air flux). The northern sub-tropical section of cruise 1 shows an overall positive $CO_2$ flux into the atmosphere, while south of 15°N the net fluxes are smaller and in variable direction. The highest magnitude fluxes were seen around the European continental shelf in cruise tracks 3 and 4, with a
strong ocean sink west of Ireland and an intermittent source of $CO_2$ in the North Sea. Figure 3b shows the difference in calculated net flux between use of the original $fCO_2$ data and the reanalysed $fCO_2$. Whilst very little difference is seen over large lengths of cruise tracks 1, 2 and 4, there are substantial differences of >50% in some regions, for example within the frontal regions at the edge of the European shelf seas (cruise track 3) or in the southern section of cruise track 1 where temporally and
spatially dynamic temperature gradients appear to exist. Interestingly, there are also examples (e.g. along the equatorial part of cruise track 1 and the western part of cruise track 2) where the direction of the flux has changed as a result of re-analysing the $fCO_2$ data.

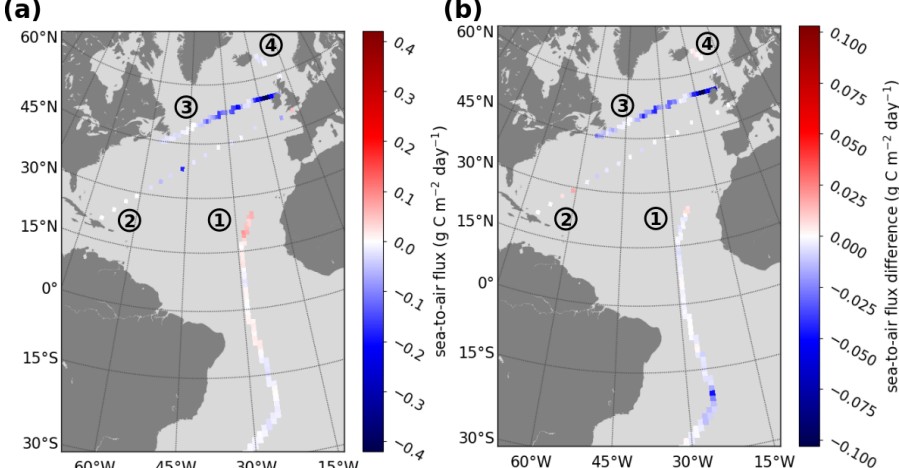

**Figure 3: Example sea-to-air $CO_2$ fluxes calculated using *in situ* data and the gas transfer velocity detailed in (Ho *et al.*, 2007) (a) fluxes calculated for four sampling cruises in the North Atlantic during October and November 2013 (Kitidis and Brown, 2017; Schuster, 2016; Steinhoff *et al.*, 2016; Wanninkhof *et al.*, 2016) labelled 1-4, respectively. (b) The difference in the calculated flux resulting from using the reanalysed $fCO_2$**



compared to the original *in situ* fCO₂ data (reanalysed minus original).


The *append2insitu.py* tool was then used to append FluxEngine output to the original input data file for the Kitidis and Brown (2017) dataset. The output from this tool enables the user to visualise FluxEngine output (including any additional input data such as the CCMP wind speed data) as a time series alongside all other measured *in situ* data. Figure 4 shows the time series of sea surface

temperature, fCO₂, and xCO₂ (from the *in situ* data) alongside the corresponding CCMP wind speed and the calculated concentrations and fluxes using the original and reanalysed fCO₂ data.

**Figure 4: Time series of the (Kitidis and Brown, 2017) *in situ* campaign data with the sea-to-air CO₂ flux as**





**calculated by FluxEngine using the Ho *et al*., (2006) gas transfer velocity parameterisation. The results from the reanalysed fCO$_2$ values are shown in red to distinguish them from the original data. The differences in fCO$_2$, sub-skin and interface CO$_2$ concentration and sea-to-air CO$_2$ flux, resulting from the reanalysis, are shown in grey (reanalysed minus original).**

**3.2 Case study 2: Calculating CO$_2$ fluxes from Östergarnsholm fixed station data**

In this section the new capabilities for calculating gas fluxes from fixed stations is demonstrated using data from the long term monitoring station at Östergarnsholm. The Östergarnsholm station is situated in the Baltic Sea (57.42N, 18.99E) and is part of the Integrated Carbon Observation System (ICOS) infrastructure. The station was originally established in 1995 with the aim of collecting data on the marine atmospheric boundary layer to support research on the exchange of heat, momentum and CO$_2$

between the atmosphere and ocean. It is equipped with instruments to measure (amongst other parameters) profiles of wind speed, water temperature and aqueous fCO$_2$.

The new FluxEngine support for calculating gas fluxes from fixed stations uses the temporal dimension of the input files, creating output files of the same dimension that can be easily visualised as a time

series. Data for the Östergarnsholm monitoring station covering a period from 28$^{th}$ January 2015 to the 9$^{th}$ September 2015 were downloaded from the data repository (Rutgersson, 2017). These data contain *in situ* measurements for fCO$_2$, salinity and temperature, model reanalysis air pressure at sea level from the National Center for Environmental Prediction, National Center for Atmospheric Research (NCEP/NCAR) dataset (Kalnay *et al*., 1996), xCO$_2$ from the NOAA ESRL GLOBALVIEW dataset

(GLOBALVIEW-CO2, 2013) and World Ocean Atlas salinity data (Boyer *et al.*, 2013). CCMP wind speed data were extracted and added to the tab delimited *in situ* dataset using the same method as used in case study 1 (Sect. 3.1). For gridded input data a single grid point containing the Östergarnsholm station location was selected from a global 1° × 1° projected grid.

The *text2ncdf.py* tool was configured to convert the text formatted data file into a single netCDF file using a temporal resolution of one day. This produced a netCDF file with a temporal dimension size of 246 (days), containing the daily mean value for each of the 246 days covered by the dataset. FluxEngine was configured to use this file as input, and to index into the temporal dimension appropriately. The fCO$_2$ data were reanalysed using the same method and data as used in case study 1

to determine fCO$_2$ at the bottom of the mass boundary layer.

The flux calculation used the rapid model (Woolf *et al*., 2016) with the Nightingale *et al.* (2000) wind based gas transfer velocity parameterisation and was performed separately using the reanalysed fCO$_2$ and original fCO$_2$ data. The temporal resolution was set to provide daily calculations for each of the

246 days allowing seasonal variations to be observed, but not diurnal variations. FluxEngine supports arbitrary temporal resolutions to within minute precision and the choice predominantly depends on the resolution of the available data and the particular research questions to be addressed. FluxEngine was configured to write output into a single netCDF file as a time series. Figure 5 shows the time series of SST, wind speed, xCO$_2$, fCO$_2$, concentration of CO$_2$ and calculated sea-to-air CO$_2$ flux. FluxEngine

output produced by using the reanalysed fCO$_2$ are plotted in red. There was a mean increase of 0.022 g C m$^{-2}$ day$^{-1}$ in sea to air CO$_2$ flux (a 35% increase in outgassing) when using the reanalysed fCO$_2$.



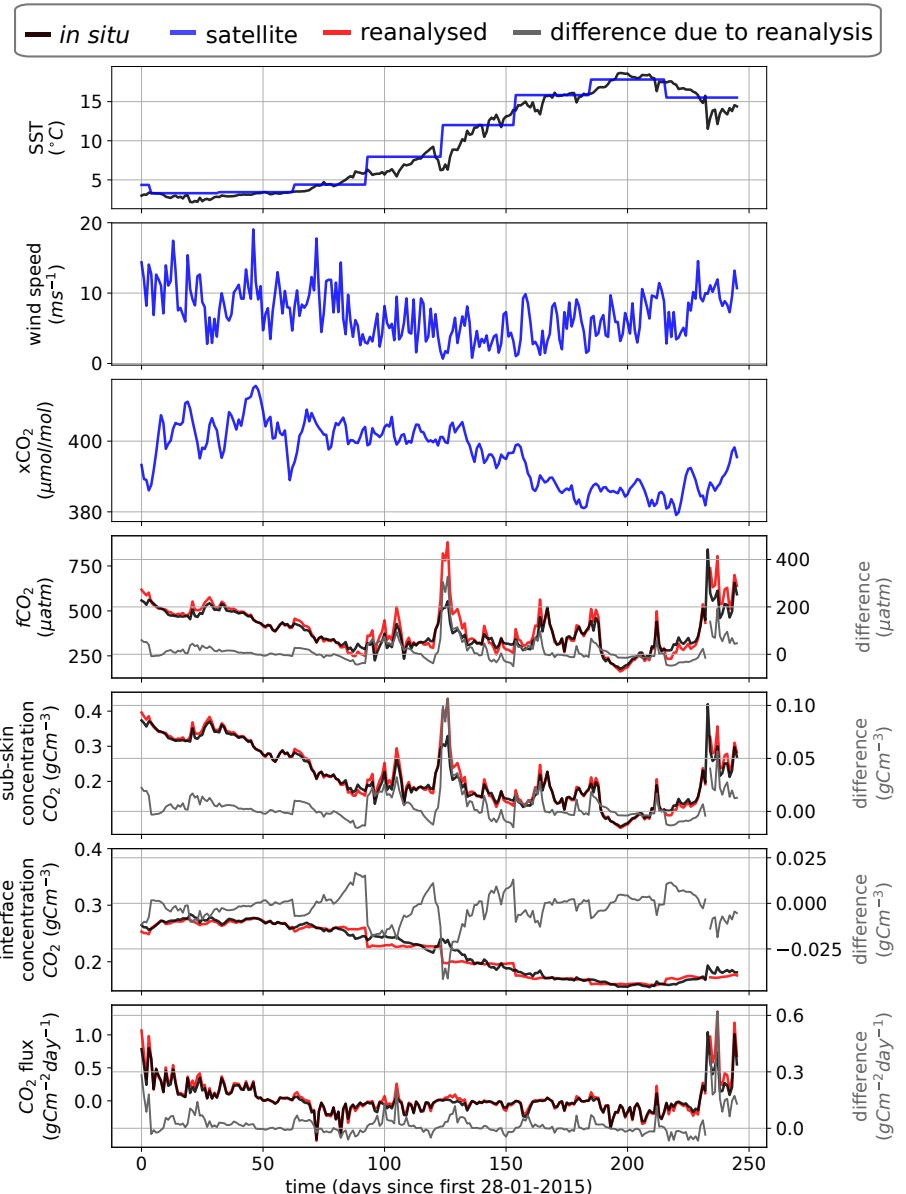

**Figure 5: FluxEngine output file using data from Östergarnsholm station over the 246 day period. Example components of the sea-to-air flux calculation are shown alongside the calculated $CO_2$ flux for $fCO_2$ reanalysed to a consistent temperature and depth and unprocessed $fCO_2$ data. The differences in $fCO_2$, subskin and interface $CO_2$ concentration and sea-to-air $CO_2$ flux, resulting from the reanalysis, are shown in grey (reanalysed minus original).**

### 3.3 Case study 3: Surfactant suppression of $N_2O$ gas fluxes using the MEMENTO database

Nitrous oxide ($N_2O$) and methane ($CH_4$) are both climatically important gases. In the troposphere, they

act as greenhouse gases (IPCC, 2013), whereas stratospheric $N_2O$ is the major source for NO radicals which are involved in one of the main ozone reaction cycles (Ravishankara *et al*., 2009). Source estimates indicate that the world's oceans play a major role in the global budget of atmospheric $N_2O$ and a minor role in the case of $CH_4$ (IPCC, 2013). Oligotrophic ocean areas are near equilibrium with



the atmosphere and, consequently, make only a relatively small contribution to overall oceanic emissions, whereas biologically productive regions (e.g., estuaries, shelf and coastal upwelling areas) appear to be responsible for the major fraction of the $N_2O$ and $CH_4$ emissions (Bakker *et al.*, 2014).

Surfactants are surface-active compounds that can suppress turbulence at the sea surface thus altering air-sea gas exchange (McKenna and Bock, 2006; Pereira *et al.*, 2016; Salter *et al.*, 2011). There is
growing evidence from field and laboratory studies that naturally occurring surfactants can significantly reduce the flux of $N_2O$ across the water/atmosphere interface (Kock *et al.*, 2012; Mesarchaki *et al.*, 2015).

Previous work, which studied $CO_2$ fluxes, found that surfactants potentially reduce the annual net
integrated $CO_2$ flux by up to 9% in the Atlantic Ocean (Pereira *et al.*, 2018). Here, we use FluxEngine to apply the methodology of Pereira *et al.,* (2018) to *in situ* data from the MEMENTO (MarinE MethanE and NiTrous Oxide) database (Kock and Bange, 2015) in order to estimate the equivalent suppression effect on the exchange of $N_2O$ between ocean and atmosphere.

While FluxEngine is able to calculate sea-to-air fluxes of both $N_2O$ and $CH_4$, we confined our analysis to $N_2O$ because of the sparsity of $CH_4$ data. *In situ* and 1° x 1° gridded monthly mean atmospheric and ocean partial pressure of $N_2O$, sea surface temperature and salinity were obtained from the MEMENTO database for the Atlantic Meridional Transect (AMT) cruise (AMT-24, JR303), which took place between September and November 2014 (Brown and Rees, 2018). These data were supplemented with
Earth observation wind speed, $U_{10,}$ from the CCMP dataset and modelled air pressure from the European Centre for Medium-Range Weather Forecasts (ECMWF). All input data were gridded to monthly means with a 1° x 1° resolution.  While sea surface temperature measured from pumped water samples collected at some depth are used here, recent AMT cruises (from 2016) have included an Infrared Sea Surface Temperature Autonomous Radiometer (ISAR) (Donlon *et al.*, 2008) and therefore
future AMT datasets will include direct sea skin temperature measurements. Using skin temperature measurements are likely to further increase the accuracy of the flux calculation.

A custom gas transfer velocity parameterisation was implemented following the template provided in the toolbox to calculate the gas transfer suppression due to biological surfactants in surface waters. This
parameterisation uses the gas transfer velocity of (Nightingale *et al.*, 2000) combined with an estimate of the degree of surfactant suppression from (Pereira *et al.*, 2018). FluxEngine was configured to use the rapid flux model (Woolf *et al.*, 2016) and run once with the standard Nightingale *et al.*, (2000) gas transfer parameterisation (no suppression case) and then again using the Pereira *et al.*, (2018) parameterisation (suppression case). This new gas transfer parameterisation is now freely available
within    the    FluxEngine    (and    can    be    selected    by    specifying *k_Nightingale2000_with_surfactant_suppression* for the *k_parameterisation* option).

The calculated sea-to-air $N_2O$ flux for each grid cell (within which at least one *in situ* measurement exists) are shown in Fig. 6a, while the difference in sea-to-air flux due to surfactant suppression is
shown in Fig. 6b. The largest fluxes in both directions occur in the tropics and sub-tropical part of the AMT cruise track (Fig. 6a). Suppression of the gas transfer reduces the magnitude of the air-sea flux


(regardless of direction of flux) and the largest absolute suppression is seen in the tropics and sub-tropical part of both (Fig. 6a and Fig. 6b).

The *append2insitu.py* utility was used to combine FluxEngine output with the original *in situ* data. The time series are shown in Fig. 6c for SST, wind speed, atmospheric and aqueous $N_2O$, and sea-to-air $N_2O$ flux. The net fluxes along the transect are generally small and in both directions. The overall mean flux was negative but small, $-2.4 \times 10^{-3} \pm 2.5 \times 10^{-2}$ g $N_2O$ m$^{-2}$ day$^{-1}$ (no suppression) and $-1.9 \times 10^{-3}$ ($\pm 2.0 \times 10^{-2}$) g $N_2O$ m$^{-2}$ day$^{-1}$ (suppression), indicating in both cases a small net flux into the ocean.

There was a mean flux suppression due to surfactants of 13% for the entire dataset, while there was an overall (net) change in flux of -20% (reducing the flux to the ocean).

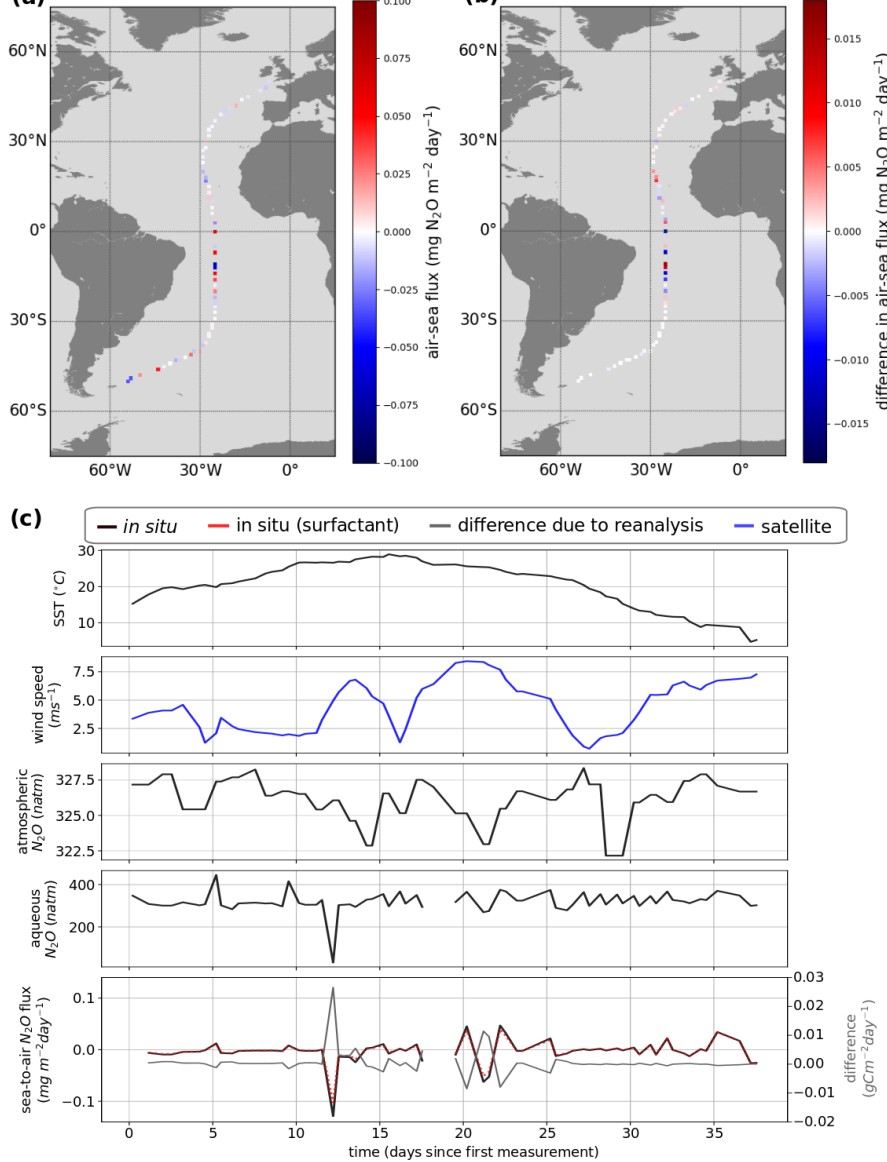




**Figure 6: (a) N₂O sea-to-air N₂O flux taking into account surfactant suppression. (b) Change in N₂O flux resulting from surfactant suppression. (c) Time series of SST, wind speed, atmospheric N₂O, aqueous N₂O and sea-to-air flux.**

### 3.4 Case study 4: Gas transfer velocity parameterisation using turbulent kinetic energy dissipation rate

The gas transfer velocity, $k$ in equation 1 and 2, is determined by the turbulent mixing near the ocean surface (Jähne *et al.*, 1987). While it is common to estimate gas transfer using a polynomial relationship with wind speed, turbulence in the upper ocean is influenced by additional physical processes which are independent or not solely dependent on the wind. These include wave breaking, shear stress due to geostrophic currents, wind-wave-current interactions, bottom-generated turbulence,

tidal forces and precipitation (Villas Boas *et al.*, 2019; Zappa *et al.*, 2007; Zhao *et al.*, 2018).

In this case study we apply a turbulent kinetic energy dissipation rate (ε) based gas transfer velocity parameterisation, as developed by Zappa *et al.* (2007), to quantify the impact of wind- and wave-driven turbulence on sea-to-air CO₂. Zappa *et al.* used direct measurements of $k$ and ε in aquatic and shallow

marine regions to derive the following relationship

$$k = 0.419Sc^{-0.5}(\varepsilon v)^{0.25} \qquad (3)$$

where $k$ is the gas transfer velocity (m s⁻¹), Sc is the Schmidt number, ε is the turbulent kinetic energy

dissipation rate (W kg⁻¹) and v is the kinematic viscosity of water (m² s⁻¹). We calculate the monthly mean ε using the monthly mean wave (swell, secondary swell and wind waves) to ocean turbulent kinetic energy flux (FOC) provided by the WAVEWATCH III model re-analysis (WAVEWATCH III development group, 2016). The mean dissipation rate of turbulent kinetic energy, $\varepsilon_{mean}$, is calculated using $\varepsilon_{mean}$ = FOC / (ρz$_{max}$), where ρ is the density of sea water (taken to be 1026 kg m⁻³) and z$_{max}$ is the

maximum depth over which dissipation is assumed to occur (taken as 10 m from Fig. 8 of Craig and Banner, 1994). This provides the mean total dissipation rate through the volume of water. Equation 3 is valid for ε measurements near the surface (of the order of 0.1 to 0.2 m) and ε is known to decrease exponentially with depth. To estimate ε at a depth of 0.2 m we first fit an exponential function to the curve of ε from fig 8 of Craig and Banner (1994) which gave:


$$\varepsilon = \beta \exp(0.20z + 0.78) \qquad (4)$$

where z is depth and $\beta$=1.86×10⁻³. Normalising this function to have a mean ε equal to $\varepsilon_{mean}$ allows ε at any depth to be determined. This was done by fitting β to minimise the difference between $\varepsilon_{mean}$

calculated from FOC and $\varepsilon_{mean}$ calculated from equation 4 to produce separate depth relationships with ε for each individual grid cell. Finally, the dissipation rate at 0.2 m was calculated by substituting z=0.2 into the final depth relationship. The process of fitting of the depth relationship and calculating ε at depth z=0.2 was implemented using a custom pre-processing function that is included as an example in the FluxEngine download. This demonstrates how pre-processing functions can be used to perform

complex data processing.




FluxEngine was then used to calculate monthly sea-to-air $CO_2$ fluxes, globally, for 2010. All inputs to FluxEngine were provided as monthly averages with a 1° x 1° resolution. The other input data were wind speed data from WAVEWATCH III re-analysis forcing field (WAVEWATCH III development group, 2016), sea surface temperature from Reynolds Optimally Interpolated Sea Surface Temperature dataset (OISST, Reynolds *et al.*, 2007), salinity data from the NOAA World Ocean Atlas (Zweng *et al.*, 2018), atmospheric molar fraction of $CO_2$ in dry air data from the GLOBALVIEW $CO_2$ dataset (GLOBALVIEW-CO2, 2013), and $fCO_2$ data from the SOCAT derived sea-to-air $CO_2$ flux reference dataset for 2010 (Woolf *et al.,* in-review). Since the Zappa et al., (2007) relationship was parameterised in low to moderate wind speeds and in shallow marine environments, a mask was set in the configuration file to constrain the calculation to grid cells with wind speeds less than 10 m s$^{-1}$ and shelf sea water depths between than 20 m and 200, and 20 and 500 m. These depth ranges were chosen to be consistent with previous studies (e.g. Laruelle et al., 2018; Shutler et al., 2016).

The *ofluxghg_flux_budgets.py* tool was used to compute the annual integrated net sea-to-air flux in all shelf sea regions. Collectively the global shelf seas result in a net integrated flux into the ocean (sink) of 0.57 to 0.78 Pg C for 2010, where the range is due to the two shelf definitions. These results are within the bounds of those determined by previous studies (0.2 – 1 PgC yr$^{-1}$ from Laruelle et al., 2018; Laruelle et al., 2016). However we note that all previous studies have used wind speed for calculating gas exchange. Repeating the analysis with a wind speed based gas transfer velocity (Wanninkhof et al., 2014) instead of equation 4 gives an ~8% smaller net integrated flux of 0.53 to 0.72 Pg C. This result could suggest that published values of the global shelf sea $CO_2$ sink (calculated using wind speed gas transfer) are underestimated, as they do not fully account for wind-wave-current interactions and whitecapping. Figure 7 shows the resulting mean annual sea-to-air $CO_2$ flux in 2010 for global shelf seas. The FluxEngine has the capability to use non-wind driven gas transfer parameterisations allowing more physically based approaches to be evaluated such as the use of ε. The first synoptic-scale observation-based estimates of ε could soon be possible from space using Doppler techniques (e.g. Ardhuin et al., 2019).

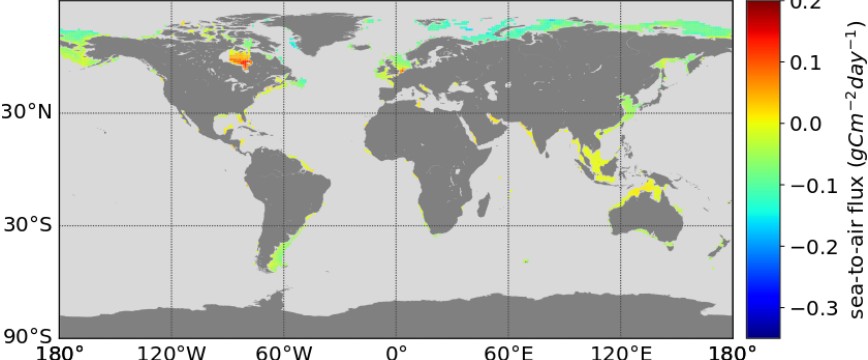

**Figure 7: Mean sea-to-air $CO_2$ flux of shelf seas in 2010 using the Zappa et a al., (2007) gas transfer relationship for all regions and months with wind speeds 0 to 10 m s$^{-1}$. Shelf regions are defined as having depth between 20 m and 200 m.**

**4. Future developments**





The FluxEngine toolbox will continue to be developed in response to new advances in research. To increase user-uptake future work will include a series of iPython Jupyter notebooks. These online and
interactive iPython notebooks will allow users to investigate the toolbox without the need to install any software. Users will be able to modify and re-run the notebook and immediately see the impact of any changes. This approach has been previously used for supporting collaborative research and summer school teaching. For example, Jupyter notebooks could be used to provide worked examples of: i) simple execution using custom input data ii) pre-processing of *in situ* data, iii) creating and testing
custom gas transfer parameterisations and pre-processing functions, iv) driving FluxEngine with a custom python script to perform a sensitivity analysis, or v) using the verification tools module to verify custom changes and extensions to the toolbox.

**5. Conclusions**

The FluxEngine is an open-source and freely available software toolbox that provides standardised and verified calculations of gas exchange and net integrated fluxes between the ocean and atmosphere, and the toolbox is now being used by *in situ*, Earth observation and modelling scientific communities. The development of the toolbox was driven by the desire to reduce duplication of effort, to facilitate collaboration between different research communities, and thus to accelerate advancements in air-sea
gas flux research and monitoring.

Building on Shutler *et al.* (2016), which demonstrated the toolbox and verified the accuracy of the calculations, this paper demonstrates new capabilities that considerably broadens the scope of research questions that can be addressed using FluxEngine. Version 3.0 can now be easily installed and
executed on a desktop or laptop computer and does not require specialist hardware or software libraries. It can be used as a python library or as a set of stand-alone command line utilities. The toolbox now includes an extensive suite of tools for calculating gas fluxes directly from *in situ* data. Collectively these improvements have streamlined the process for extending the toolbox and will allow users to easily take advantage of newly developed gas transfer velocity parameterisations and/or new
sources of input data. These new tools and the toolbox are fully compatible with the internationally agreed data structures being used by the SOCAT and the MEMENTO communities.

The inclusion of the handling of $CH_4$ and $N_2O$ sea-air gas fluxes is intended to directly support those communities studying these gases. Significant international research focus and effort is now being
directed to collating data on these gases towards monitoring and understanding their spatial distribution and variability.

FluxEngine will continue to be updated as new approaches become available. Further development will be guided by the needs of the international research and monitoring communities, and so we welcome
feedback from users on all aspects of the toolbox.

**Code availability**

The FluxEngine software is open source and available on a creative commons license via http://github.com/oceanflux-ghg/FluxEngine





### 6. Appendix A: Utility names and descriptions

Several additional utilities are provided as Python scripts to support the installation, verification, execution and processing of output (these are listed in table 2).

| Utility | Description |
|---------|-------------|
| *append2insitu.py* | Appends netCDF data (e.g. FluxEngine output) to text formatted data files as new columns. Matching rows by longitude, latitude and time. |
| *install_dependencies_macos.py,* *install_dependencies_ubuntu.py* | Installation scripts. Installation instructions are provided for Windows users in *FluxEngineV3_instructions.pdf* |
| *ncdf2text.py* | Converts netCDF output files to text formatted files. |
| *ofluxghg_flux_budgets.py* | Calculates total monthly and annual gas flux from FluxEngine output. Supports global and regional analysis. |
| *ofluxghg_run.py* | Commandline tool used to run FluxEngine |
| *reanalyse_socat_driver.py* | Uses satellite sea surface temperature to reanalyse $CO_2$ fugacity and partial pressure data to a consistent temperature and depth (see Goddijn-Murphy *et al.,* 2015) |
| *run_full_verification.py* | Runs an extended verification procedure. Required additional data from (Holding *et al.*, 2018) |
| *text2ncdf.py* | Converts text formatted data files into FluxEngine compatible netCDF format. |
| *validation_tools.py, compare_net_budgets.py* | Contains Python functions to aid verification of FluxEngine output to a reference dataset. |
| *verify_socatv4_sst_salinity_gradients_N00.py,* *verify_takahashi09.py* | Verifies that FluxEngine has been installed correctly by comparing output with a reference data from SOCAT-derived or Takahashi climatologies, respectively. |

**Table 2:** Description of the bundled tools and scripts that are included in FluxEngine. Each tool can be used as a stand-alone command line tool or used as a Python package.

### 7. Appendix B: Datasets used

Table 3 provides details of each of the data sets that were used in the case studies.

| Name | Parameter(s) | Reference/source |
|------|--------------|------------------|
| CCMP v2 (Cross-Calibrated Multi-Platform) | $U_{10}$ (wind speed at 10m) | Atlas *et al.*, 2011 http://www.remss.com/measurements/ccmp / |
| OISST (Optimally-Interpolated Seas Surface Temperature) | Sea surface temperature (SST) | Reynolds *et al.*, 2007 https://www.ncdc.noaa.gov/oisst |
| GLOBALVIEW $CO_2$ | $xCO_2$ (molar fraction of $CO_2$ in dry air) | GLOBALVIEW-CO2, 2013 https://www.esrl.noaa.gov/gmd/ccgg/globalview/co2/co2_intro.html |
| National Centers for Environmental Prediction, National Center for | Air pressure | Kalnay *et al.*, 1996 https://www.esrl.noaa.gov/psd/data/gridded/data.ncep.reanalysis.pressure.html |




| | | |
|---|---|---|
| Atmospheric Research (NCEP/NCAR) | | |
| Underway data from the James Clark cruise (74JC20131009) | SST, salinity, air pressure, $fCO_2$ | Kitidis and Brown, 2017 https://doi.pangaea.de/10.1594/PANGAEA.878492 |
| Underway data from the Belguela Stream cruise (642B20131005) | SST, salinity, air pressure, $fCO_2$ | Schuster, 2016 https://doi.org/10.1594/PANGAEA.852980 |
| Underway data from the Atlantic Companion cruise (77CN20131004) | SST, salinity, air pressure, $fCO_2$ | Steinhoff *et al.*, 2016 https://doi.org/10.1594/PANGAEA.852786 |
| Underway data from the REYJAFOSS cruise (64RJ20131017) | SST, salinity, air pressure, $fCO_2$, $xCO_2$ | Wanninkhof *et al.*, 2016 https://doi.org/10.1594/PANGAEA.866092 |
| Östergarnsholm station (77FS20150128) | Air pressure, salinity, SST, $xCO_2$ (air), $fCO_2$ (water) | Rutgersson, 2017 https://doi.pangaea.de/10.1594/PANGAEA.878531 |
| MarinE MethanE and NiTrous Oxide database (MEMENTO) | SST, $pN_2O_{air}$, $pN_2O_{water}$, | Kock and Bange, 2015 https://memento.geomar.de/ |
| National Oceanic and Atmospheric Administration, US (NOAA) WAVEWATCH III | $U_{10}$ (wind speed at 10m), FOC (wave to turbulent kinetic energy) | WAVEWATCH III development group, 2016 |

Table 3: The Earth observation *in situ*, model and climatology data used in this research.

**Author contributions**

Design and analysis performed by T. Holding, I. Ashton and J. Shutler. Software engineering performed by T. Holding and I. Ashton. Pre-processing of nitrous oxide data performed by A. Kock. All authors contributed to the preparation of the manuscript.

**Acknowledgements**

This work was partially funded by the European Space Agency (ESA) Support to Science Element (STSE) through the OceanFlux Greenhouse Gases project (contract 4000104762/11/I-AM), the OceanFlux Greenhouse Gases Evolution project (contract 4000112091/14/I-LG), and by the European Space Agency (ESA) through the Sea surface Kinematics Multiscale monitoring (SKIM) Mission Science study (contract 4000124734/18/NL/CT/gp) and the ESA SKIM Scientific Performance

Evaluation study (contract 4000124521/18/NL/CT/gp), as well as through the NERC RAGNARoCC project, (grant ref. NE/K002473/1). Further development of FluxEngine was funded by the European Union's Seventh Programme for Research and Technology Development (grant no. 03FO773A (BONUS INTEGRAL) and grant no. 730944 (RINGO)).



The Surface Ocean $CO_2$ Atlas (SOCAT) is an international effort, endorsed by the International Ocean Carbon Coordination Project (IOCCP), the Surface Ocean Lower Atmosphere Study (SOLAS) and the Integrated Marine Biosphere Research (IMBeR) program, to deliver a uniformly quality-controlled surface ocean $CO_2$ database. The many researchers and funding agencies responsible for the collection of data and quality control are thanked for their contributions to SOCAT.


CCMP Version-2.0 vector wind analyses are produced by Remote Sensing Systems (http://www.remss.com). NCEP Reanalysis data provided by the NOAA/OAR/ESRL PSD, Boulder, Colorado, USA, from their web site at https://www.esrl.noaa.gov/psd/.

MEMENTO (https://memento.geomar.de/) is currently supported by the Kiel Data Management Team at GEOMAR and the BONUS INTEGRAL Project.

This study is a contribution to the international IMBeR project and was supported by the UK Natural Environment Research Council National Capability (CLASS Theme 1.2) funding to Plymouth Marine
Laboratory. This is contribution number 330 of the AMT programme.

BONUS INTEGRAL receives funding from BONUS (Art 185), funded jointly by the EU, the German Federal Ministry of Education and Research, the Swedish Research Council Formas, the Academy of Finland, the Polish National Centre for Research and Development, and the Estonian Research Council.

**Competing interests**

The authors declare that they have no conflict of interest.

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
