# Peer review of "The FluxEngine air-sea gas flux toolbox: simplified interface and extensions for *in situ* analyses and multiple sparingly soluble gases"

_Ocean Science, 2019_

## Referee Comment (RC1) · Anonymous Referee #1 · 17 Jul 2019

This is a toolbox for calculating gas exchange fluxes and perform related calculations. The manuscript describes new capabilities of the toolbox and walks the reader through examples of the toolbox in use and some of its capabilities. The case examples are helpful as they contain imperfect data and are therefore more useful than simply denoting the computational equations. The new version of the toolbox appears like a useful extension that seems targeted towards improving the ways in which it can be used and making sure the data formats are compatible with the larger research community. My only minor comment is that including a table describing the overall capabilities of the FluxEngine toolbox would be useful to potential users who are not familiar with the tool. Listing the gases, air-sea flux parameterizations, most common adjustments (skin

layer, constant temperature, etc) would help the reader to know whether it is worth their time and effort to learn to use the toolbox. As it is currently written the paper assumes readers have a baseline understanding of what is contained within the FluxEngine toolbox. Overall this manuscript provides clear examples of the type of calculations contained within the FluxEngine toolbox and seems like a useful tool to have in the community. I recommended it be accepted with this one minor change.
* * *

---

## Referee Comment (RC2) · Brian Butterworth (Referee) · 22 Jul 2019

=============== General Comments ===============

This manuscript describes the new functionality built into the FluxEngine toolbox for calculating bulk fluxes of air-sea gas exchange. These features include the ability to calculate fluxes for additional sparingly soluble gases (e.g., N2O and CH4), as well as additional support for in situ data analyses, such as user defined grids, time periods, projections, and the ability to reanalyze CO2 data during the data ingestion procedures. These improvements make the toolbox a more useful tool for the scientific community. The example case studies provided here give the reader context for its usefulness,

giving some concrete examples of how the community can apply the toolbox. The manuscript is effective in its description of the new features of FluxEngine. The authors provided example configuration files for each of the four case studies presented, which was helpful for following along with how to apply the new features. I have some minor comments. Besides that, I recommend the article for publication.

One possible concern with this manuscript is its shelf life. Unlike previous types of bulk flux scripts (e.g., NOAA COARE) which had unique versions that did not change after publication, this toolbox will presumably continue to change. That may have the effect of causing this article to be difficult to follow even a year or two from now. It would be helpful for someone reading this article in the future to be able to view a version of the toolbox from the time the article was written. I believe that is a function of Git, but I'm not sure how one would accomplish it. If it is possible, it may be worth adding the steps to the instructions pdf and then mentioning it in the main text. That way someone can find the items being described in this paper after they've been modified or removed from the toolbox.

=============== Specific Comments ===============

Line 45: Add a link to the toolbox in the abstract (if that doesn't violate editorial rules). You'll get more clicks.

Lines 92-93: You mention that there is an option of either bulk formula, equation 1, or equation 2. It might be useful to include the bulk formula as an equation in the paper. The two equations that you do show are different iterations of the bulk formula. So, I'm not sure how the bulk formula option differs.

Line 321: Include link to pangaea.de (my top Google results were for Oklahoma oil and gas data at pangaeadata.com)

Line 345: The text says the intakes were of unknown depth. When doing the reanalysis what depth was entered?

Line 384: Cruise 4 does not look like its flux magnitudes were much higher than 1 and 2.

Figure 4: Color of xCO2 is blue, but text says that it's in situ. Shouldn't it be black?

Figure 3 & 4: One says gas transfer velocity was from Ho et al. (2006), the other Ho et al. (2007). Is that correct?

Line 429: The text says that the same method was used to reanalyze Case Study 2 and Case Study 1, but the monthly satellite SST used to reanalyze the SOCAT datasets (seen in Fig. 4a) appears to be a moving average to higher temporal resolution, while Ostergarnsholm was stepped monthly SST values (Fig. 5z). Why the difference?

On the topic of reanalyzed fluxes... I understand that the purpose of this paper is to highlight the functionality of FluxEngine, but as long as you're showing data plots it would be good to have a better description of their importance. It doesn't seem likely that the monthly satellite SST is more accurate than high temporal resolution in situ SST measurements obtained at a non-standardized depth. I could be wrong. But if I am it would be good to make that clear. Because that difference appears to be a major driver in CO2 flux difference between original data and reanalyzed data. So, then statistics, such as 35% difference between original and reanalyzed fluxes, don't mean very much. It just feels like an exercise. Again, I understand that the science questions are not the purpose of the paper. But the examples would be stronger, and more engaging, if it seemed like the differences being shown were indicative of true error in the in situ measurements.

Line 481: The text mentions that Pereira et al. (2018) was used to estimate the degree of surfactant suppression. But what data was used to estimate where surfactants were physically present? Doesn't there still need to be some underlying data layer? Or is the Pereira estimate a blanket effect for all grid cells?

Line 494: I am not sure to what "parts of both" refers. Was it both figures?

Line 552: What dataset did you use for sea water depths?

Figure 7 caption: Add "annual" after "Mean"

=============== Technical Corrections ===============

Line 28: Consider changing "our" to "the" (e.g., health of the oceans and changes to the climate). It does not need to be personal.

Line 43: "supress" to "suppress"

Lines 53 - 54: SOCAT and MEMENTO citations don't match in format – one has parentheses the other doesn't.

Line 100: Remove comma after "Shutler"

Line 106: Add comma after "At the time of writing"

Line 120: "as described by" is unnecessary

Line 125: Comma after "Collectively"

Line 328: Comma after "Therefore"

Line 340: Capitalize python

Line 346: Add "on" before "either side"

Line 349: Add comma after "Therefore"

Line 359: Add the following commas: "Here, for simplicity,"

Line 437: Consider starting the sentence that begins "FluxEngine was configured to write output . . ." with "For this case study,"

Line 529: "fig 8" to "Fig. 8" or "Figure 8"

Line 556: Add comma after "Collectively"

Line 596: Capitalize python

Line 598: Add comma after "Collectively"

Line 601: Change "agreed" to "agreed-upon" or "established" or similar

Figure 3 caption: change "detailed in (Ho et al. 2007)" to "detailed in Ho et al. (2007)"

---

## Author Comment (AC1) · 23 Oct 2019

The referee's comments are shown below with our reply and action shown in **bold**. Quoted line numbers refer to the line numbers in the tracked changes document.

**Responses to Reviewer 1**

Reviewer general comments
My only minor comment is that including a table describing the overall capabilities of the FluxEngine toolbox would be useful to potential users who are not familiar with the tool. Listing the gases, air-sea flux parameterizations, most common adjustments (skin layer, constant temperature, etc) would help the reader to know whether it is worth their time and effort to learn to use the toolbox. As it is currently written the paper assumes readers have a baseline understanding of what is contained within the FluxEngine toolbox.

**We agree. A table describing the overall capabilities and commonly used options available would aid potential users. We have added this table into section 8 (Appendix C) of the manuscript and the table is highlighted within the introduction section of the paper on lines 219 to 220.**

---

## Author Comment (AC2) · 23 Oct 2019

The referee's comments are shown below with our reply and action shown in **bold**. Quoted line numbers refer to the line numbers in the tracked changes document.

**Responses to Reviewer 2**

We thank Dr Butterworth for his insightful comments and thorough review of the manuscript.

Reviewer general comments

One possible concern with this manuscript is its shelf life. Unlike previous types of bulk flux scripts (e.g., NOAA COARE) which had unique versions that did not change after publication, this toolbox will presumably continue to change. That may have the effect of causing this article to be difficult to follow even a year or two from now. It would be helpful for someone reading this article in the future to be able to view a version of the toolbox from the time the article was written. I believe that is a function of Git, but I'm not sure how one would accomplish it. If it is possible, it may be worth adding the steps to the instructions pdf and then mentioning it in the main text. That way someone can find the items being described in this paper after they've been modified or removed from the toolbox.

**The reviewer raises an important point. While we would discourage users from using old versions of the toolbox that may contain old methodology, we recognise the critical importance of transparency and repeatability in science and that part of this involves being able to trace and reference specific versions of scientific tools. Historic versions of FluxEngine are accessible through the Github page and we have frozen development of version 3.0 with the submission of this manuscript. This version is relevant for this manuscript and it is now permanently accessible through the Github repository's releases page: https://github.com/oceanflux-ghg/FluxEngine/releases. A link to this has been added to the manuscript. We have expanded the section on code availability to explain this (please see lines 987 to 989).**

**In addition, the FluxEngine repository includes the configuration files used for each case study as examples to aid new users in constructing their own configuration files. These will continue to be updated to maintain their compatibility with future releases. To help, we have also added four interactive Jupyter tutorials that are based on the first three case studies from our manuscript. These tutorials are included in the FluxEngine download and provide all of the information, data and code required to reproduce the case studies. It is our intention to include these Jupyter tutorials in all future releases of the toolbox so that the software remains relevant to the content in the paper.**

**We have now added a statement into the discussion to introduce the tutorials and to explain our intention to keep these within future releases (please see lines 501 to 504 and 929 to 953).**

Line 45: Add a link to the toolbox in the abstract (if that doesn't violate editorial rules). You'll get more clicks.

**The guidelines for authors asks that citations not be included in abstracts unless absolutely necessary. As such we have refrained from including a link to the toolbox's Github page here. Instead, we have added a link into the introduction of the paper. Please see line 75.**

Lines 92-93: You mention that there is an option of either bulk formula, equation 1, or equation 2. It might be useful to include the bulk formula as an equation in the paper. The two equations that you do show are different iterations of the bulk formula. So, I'm not sure how the bulk formula option differs.

**The equations shown, as pointed out by the reviewer, are formulations of the bulk equation, but utilise different solubility terms for the atmospheric and aqueous component. The focus of the current manuscript is on the new features provided by FluxEngine and so rather than repeating previous discussions of the theory we have added a sentence to refer readers to the relevant literature (Woolf et al. 2016 and Shutler et al. 2016) and we have also clarified the explanation of the equation so we now refer to these equations as variations of a bulk formula. Please see the modifications to lines 79 to 85.**

**One reason for our original choice to give the more accurate bulk formulations in the manuscript (e.g. our choice to state equations 1 and 2) is to encourage the scientific community to use these more accurate formulations, as the commonly used approximation (using $\Delta pCO_2$ and containing only one solubility term) can result in substantial biases. This issue is discussed in detail within Woolf et al. (2016).**

Line 321: Include link to pangaea.de (my top Google results were for Oklahoma oil and gas data at pangaeadata.com)

**Done. This is now given on line 543. The link to the dataset that this is referring to (Holding et al., 2019) is also provided in the reference list.**

Line 345: The text says the intakes were of unknown depth. When doing the reanalysis what depth was entered?

**The reanalysis step does not require depth to be explicitly defined. Instead it uses a temperature dataset that is referenced to known depth. The details of the method can be found in Goddijn-Murphy et al., 11(4), Ocean Science, 2015. This published method simply requires a temperature that is valid for a consistent depth in all ocean regions, so we have used a satellite observed and climate quality dataset and in our analysis this is used to represent the temperature at the base of the mass boundary layer. The method relies on the paired temperature and fugacity ($fCO_2$) measurements. So the $fCO_2$ is recalculated based on the difference in temperatures between the *in situ* temperature measurement (which was collected at some unknown depth) and depth-consistent (satellite observed) temperature field to produce the $fCO_2$ values that are consistent**

with the satellite observed temperature (and therefore valid for a consistent depth).

**The satellite observed temperature dataset used within the case studies are valid for a depth of ~1 m (Reynolds et al. 2007). So the re-calculated fCO$_2$ are therefore also valid for this depth. We have added this statement into the paper on lines 363 to 366.**

**We have expanded the explanation of these issues. Please see lines 342 to 427.**

Line 384: Cruise 4 does not look like its flux magnitudes were much higher than 1 and 2.
**Apologies and thank you for highlighting this. This sentence has been corrected to just refer to cruise track 3. Please see line 659.**

Figure 4: Color of xCO2 is blue, but text says that it's in situ. Shouldn't it be black?
**The xCO$_2$ data were acquired as part of the downloaded cruise data, but the data documentation indicates that they are actually interpolated from the GLOBALVIEW-CO2 dataset, and so it is therefore not *in situ*. We have amended the text to clarify this. Please see lines 549 to 550.**

Figure 3 & 4: One says gas transfer velocity was from Ho et al. (2006), the other Ho et al. (2007). Is that correct?
**Apologies. This was in error. The captions for figures 3 and 4 now both say Ho et al. 2006.**

Line 429: The text says that the same method was used to reanalyze Case Study 2 and Case Study 1, but the monthly satellite SST used to reanalyze the SOCAT datasets (seen in Fig. 4a) appears to be a moving average to higher temporal resolution, while Ostergarnsholm was stepped monthly SST values (Fig. 5z). Why the difference?
**Both case studies did originally use the same method. The monthly 'stepping' is visible in the Ostergarnsholm fixed station data but not in the cruise data for two reasons:**
**1) The spatial resolution of the temperature data is 1° by 1° and as the research vessel moves across grid boundaries this results in different temperature values within the same month. Whereas the Ostergarnsholm data are at a fixed location (57.42N, 18.99E), so the monthly mean temperature remains constant throughout the month.**
**2) The research cruise shown in Figure 4 takes approximately 30 days (starting on the 16$^{th}$ October, 2013) and so this period overlaps two months. In contrast, the Ostergarnsholm data in Figure 5 spans ~250 days.**

**These differences in time and space of the two case studies results in differing SST variability within the plots in figure 4 and figure 5.**

**However, this comment from the reviewer and an observation by one of co-authors has meant that we have updated the Ostergarnsholm fixed station**

**analysis to omit the re-analysis as we feel that its application was misleading. So the original 'stepped' issue that the reviewer commented on is no longer in the updated manuscript.**

On the topic of reanalyzed fluxes. . . I understand that the purpose of this paper is to highlight the functionality of FluxEngine, but as long as you're showing data plots it would be good to have a better description of their importance. It doesn't seem likely that the monthly satellite SST is more accurate than high temporal resolution in situ SST measurements obtained at a non-standardized depth. I could be wrong. But if I am it would be good to make that clear. Because that difference appears to be a major driver in CO2 flux difference between original data and reanalyzed data. So, then statistics, such as 35% difference between original and reanalyzed fluxes, don't mean very much. It just feels like an exercise. Again, I understand that the science questions are not the purpose of the paper. But the examples would be stronger, and more engaging, if it seemed like the differences being shown were indicative of true error in the in situ measurements.

**We agree with the reviewer that it is important to more fully explain the need for the steps taken in the example analyses. To this end, we have added a new section (section 2.4) that contains an expanded explanation of the need for the reanalysis. This is an overview of the main issues as we still refer the reader to the original publications of the full explanations and justifications.**

**It was not our intention to imply that the *in situ* SST data are less accurate than the monthly satellite SST for the measurement time and location (and depth). Instead we were attempting to explain that the paired *in situ* $fCO_2$ and SST data are collected at an unknown (and potentially variable) depth below the surface (e.g. 1 m or more). Whereas for an accurate gas flux calculation values of SST and corresponding $fCO_2$ need to be available for the bottom and top of the mass boundary layer (e.g. either side of the top 1 mm of the water-air interface). The theory and reasoning is explained within Woolf et al. (2016). So for an accurate calculation, some sort of re-analysis step is required to determine an $fCO_2$ and SST pairing that are representative of the conditions at a fixed depth that is close to the air-water interface, which can then in turn be used to represent the bottom of the mass boundary layer. The $fCO_2$ and SST at the top of the mass boundary layer can then be estimated (or vice versa).**

**This re-analysis to a consistent depth then in turn allows a more accurate calculation of the gas fluxes, as it is then possible to calculate two solubilities and thus two concentrations (one at the bottom, and one at the top of the mass boundary layer).**

**The re-analysis step reduces uncertainty and unknown biases that arise due to the $fCO_2$ measurements being collected at some unknown (and potentially variable) depth below the surface. This depth of a few metres is non-optimal for representing the bottom of the mass boundary layer and**

**could vary within an individual cruise dataset, e.g. the depth of underway samples will vary with sea state and ballasting.**

**The choice of reference SST data set to use with the reanalysis tool depends, to some extent, on the aims of the analysis. If FluxEngine is being used with a collated data set to calculate temporally averaged fluxes (e.g. monthly mean values), then using a monthly gridded SST is preferable because this provides a SST data at a consistent depth and avoids issues of sparse sampling. Alternatively, if FluxEngine is being used to calculate fluxes along a specific cruise track (or a single location, as in case study three), the best solution would be collect *in situ* sea skin temperature data and then perform the reanalysis using these data. However, these data were not normally available as most ships collecting fCO$_2$ data do not collect skin temperature (but instruments to make this measurement are available e.g the Infrared Sea surface temperature Autonomous Radiometer (ISAR). This is approach is highlighted in the new section 2.4 (lines 402 to 416).**

**We have now explained these reasoning for the re-analysis steps and assumption in more detail within Section 2.4 (lines 342 to 427) and we thank the reviewer for highlighting the need to include this explanation.**

Line 481: The text mentions that Pereira et al. (2018) was used to estimate the degree of surfactant suppression. But what data was used to estimate where surfactants were physically present? Doesn't there still need to be some underlying data layer? Or is the Pereira estimate a blanket effect for all grid cells?
**The published Pereira et al. (2018) method for estimating surfactants coverage is a linear relationship with sea surface temperature. So the temperature field provides an estimate of the surfactant coverage or its existence. The justification and reasoning for this parameterisation is contained within the original Pereira et al. (2018) paper. We have now added a sentence to clarify that no additional data were required. Please see lines 804 to 806.**

Line 494: I am not sure to what "parts of both" refers. Was it both figures?
**Yes, this is correct. We have modified this sentence to clarify the meaning. Please see line 816.**

Line 552: What dataset did you use for sea water depths?
**The GEBCO Digital Atlas bathymetry was used.**
**We have added this reference to the manuscript. Please see line 907 to 909. Apologies as this this was previously missing.**

Figure 7 caption: Add "annual" after "Mean"
**Done.**

Technical corrections
We thank the reviewer again for their detailed reading of the manuscript. We have implemented all of the suggested technical corrections.

**Additional amendments made by the authors**
Linked to Reviewer 2's comments and discussions between co-authors we have added a cautionary note that the reanalysis method assumes isochemical conditions (lines 395 to 400), and placed a greater emphasis on our recommendation that, in the ideal case, both skin and bulk ocean temperature be measured *in situ*. We decided to remove the reanalysis step from case study two as this (as Reviewer 2 identified) could cause confusion. This region is known to exhibit up-welling events and so it violates the isochemical assumptions.

The reanalysis tool is still demonstrated in case study one, and we have updated the description of the change in calculated net flux due to applying reanalysis to quote values in C m$^{-2}$ day$^{-2}$ rather than percentage difference (line 663). We feel that this is a more representative description, because the largest percentage changes occur when the magnitude of flux is small, and is therefore of little consequence.

---

## Author Response (AR2)

Dear Editor,

We are grateful to for the thorough comments and helpful suggestions. We have responded in full to all comments, with the original comment show below and out reply and action shown in **bold**. Quoted line numbers refer to the line numbers in the tracked changes manuscript (appended to the bottom of this document).

Yours sincerely,
Tom Holding and co-authors.

**Response to Mario Hoppema's comments**

General Comments
My general comment to the manuscript is that in some places, the toolbox is kind of described as if the authors would like to sell something. Also, it sometimes feels as if reading a manual. I realize that the subject of the paper sometimes rather coerces to such writing, but possibly the authors can avoid it to some extent. Please go through the text and consider where such text can be improved.
**All text has been reviewed for improvements (changes throughout the text). We have removed several links to the Github page and consolidated this information in 6 (Code Availability). Section 1 has been made more concise by considerably shortening the description of previous and on-going projects that utilise FluxEngine.**

Specific comments
L26 „the biogeochemical development" I find this a strange combination. Any better wording?
**Thank you, we have updated this sentence (line 26-27).**

L27-28 "to monitor the health of the oceans" I think speaking of health in this context is not scientifically objective, and not necessary here. Please use other wording.
**This sentence has been re-worded to be more explicit in our meaning (lines 28-29).**

L31 delete: and the toolbox is already being used by scientists across multiple disciplines (this is not info for a scientific paper)
**Done (line 31)**

L74 Shutler et al. (2016) and Woolf et al. (2016). (format)
**Done (line 78)**

L76 … by Woolf et al. (2016) and … (format)
**Done (line 80).**

L86-87 (e.g., Ho et al., 2006; Nightingale et al., 2000; Wanninkhof, 2014). (format)
**Done (lines 98-99)**

L110-111 "and broadened the range of possible applications to which it can be applied." Delete "to which it can be applied" because this does not add any info.
**Done (line 122).**

L165 Suggested: ... with the reference data published in Holding et al. (2018).
**Done, thank you (Appendix D, line 880).**

§2.1 While I was reading this, it felt like reading a manual. Although this part of the manuscript contains important information, I think a large part should be transferred to the supplementary material or appendix, as this is not the kind of info for a scientific paper. Please consider how best to do this.
**The technical details have been removed and placed in Appendix D. Section 2.1 now gives a short overview, with reader being directed to Section 10 (Appendix D) for the specific details on the location of installation scripts, the verification options (and datasets used) and benchmarking information.**

L273 Goddijn-Murphy (typo)
**Thank you, this has been corrected (line 375).**

L354-356 "It is advisable to always use the most up-to-date version of FluxEngine which can be found via http://github.com/oceanfluxghg/FluxEngine." This is not a sentence that belongs in a scientific paper.
**We have removed this sentence as well as the link to the releases page. Instead we directed the reader to the code availability section (sect. 6) (lines 483-484).**

L464-465 from 28 January 2015 to 9 September 2015 (format)
**Done (line 607).**

Fig.5 The units of xCO2 and fCO2 contain strange symbols, please correct.
**Thank you. This has been corrected.**

L531 ... of Nightingale et al. (2000) ... (format)
**Done (line 677)**

L532 from Pereira et al. (2018). (format)
**Done (line 679)**

L622 Zappa et al. (2007) (typo)
**Done (line 718)**

L685 Datasets used: SOCAT is not listed here. Is there any reason for that?

**While the individual SOCAT cruises uses in case study one were listed, the entry for the interpolated collated SOCAT dataset used in case study 4 was missing. We have now added references to the methodology used to produce this dataset, the dataset download and original SOCAT dataset (see the last row of table 3, Appendix B).**

L831 pages: 1937-1949
**Done (line 1034).**

[revised manuscript text omitted]